# A Novel Tightly Coupled Information System for Research Data Management

Kennedy Senagi *[ID] and Henri E. Z. Tonnang

International Centre of Insect Physiology and Ecology, Nairobi 30772-00100, Kenya
* Correspondence: ksenagi@icipe.org

**Abstract:** Most research projects are data driven. However, many organizations lack proper information systems (IS) for managing data, that is, planning, collecting, analyzing, storing, archiving, and sharing for use and re-use. Many research institutions have disparate and fragmented data that make it difficult to uphold the FAIR (findable, accessible, interoperable, and reusable) data management principles. At the same time, there is minimal practice of open and reproducible science. To solve these challenges, we designed and implemented an IS architecture for research data management. Through it, we have a centralized platform for research data management. The IS has several software components that are configured and unified to communicate and share data. The software components are, namely, common ontology, data management plan, data collectors, and the data warehouse. Results show that the IS components have gained global traction, 56.3% of the total web hits came from news users, and 259 projects had metadata (and 17 of those also had data resources). Moreover, the IS aligned the institution's scientific data resources to universal standards such as the FAIR principles of data management and at the same time showcased open data, open science, and reproducible science. Ultimately, the architecture can be adopted by other organizations to manage research data.

**Keywords:** information systems; research data; data management; data engineering; software engineering; common data model



## 1. Introduction

New technologies frontier toward data augmentation, data availability, digitized data collection mechanisms (e.g., using internet of things), and improved communication channels. These ideas open the space for enabling unprecedented possibilities for informing and transforming the globe. In the current world of data and big data (i.e., data that is of large volume, have a variety of data types and streams with high velocity), flourishing companies, governments, institutions, research agencies, etc., have leveraged their core business by adapting new pathways and innovations centered around data and IS. Therefore, organizations must embrace cutting-edge technologies and innovations to stay afloat in this era of data revolution [1].

Kanza and Knight [2] stated that, "Behind every great research project there is great data management". Most research projects are data driven, and proper digital data management is one of the main pillars of a successful research project. However, even when required by funding bodies, efficient digital data management often remains an underappreciated art that is overlooked in day-to-day project management activities. It is therefore imperative to plan and set research data management strategies early in research project works by coming up with a comprehensive data management plan that outlines how data are collected, stored, published, and shared, while at the same time adhering to ethics, FAIR principles of data management, reproducible, and data standards [2]. Besides these, it is also necessary to put in place efficient and robust research data management IS and skilled personnel to collect high-quality data that will lead to accurate analysis, data accessibility,

sharing, and re-use of data [3]. An IS provides a mechanism for organizing information so that an organization can achieve specific goals, such as central data management, improve customer service, increase in profits, increase in production, etc. [4]. In essence, data management involves collecting, annotating, and maintaining high-quality data that are fit for a specific purpose, e.g., statistical modeling, and machine learning.

Data should also conform to the FAIR (i.e, Findable, Accessible, Interoperable, Reusable) data management principles. Findable—both humans and machines should easily search and discover data; accessible—data archived in long-term public/private storage areas can be made available; interoperable—data can be shared and consumed across different applications and systems. This can be made possible through the use of appropriate metadata and standardized ontology/vocabulary; Re-usable—data are well documented, curated, and provide rich information about the context of data creation. When re-using data users can reference the license, access the data, and validate the results (i.e, data reproducibility) as well as deciding to use the data in a similar or different context [5].

Data quality control checks (QCCs) are part of the broad data management practice. It describes the degree to which a given piece of data fits a particular purpose and should at least be accurate, timely, complete, and consistent. QCCs consume up to 80% of a data scientist's day-to-day work. High-quality data make it possible to provide accurate data-driven solutions and decision-making at the right time. Generally, a good IS is a key conduit to data management and innovation [5–8].

In the process of fulfilling its mandate to enabling people living in the tropics and the world at large to overcome poverty, the International Center of Insect Physiology and Ecology (*icipe*) has dealt with many types of data and knowledge in the domain of insect science since its inception in the 1970s. Since then, various research programs and projects resident at and collaborating with the institution have generated and are still generating considerable amounts of data from diverse domain knowledge, such as entomology, agronomy, biology, chemistry, environmental science, geography, etc. Different forms of data are collected include laboratory/experiment notes, genomic data, household data, field notes and journals, photographs, geo-spatial, video and audiotapes, statistical package output files, technical reports, simulation, derived data, publications, etc. These data are processed (using various techniques that are relevant to those fields of study and analyzed to provide different insights and knowledge that impact humanity in diverse ways [9].

However, most of the scientific data and processes are not digitized and centralized for ease of management. Data are also distributed among individual scientists, projects, and partner institutions and organizations, which makes it difficult for others (within or outside the organization) to locate, access and possibly re-use. So far, research outputs have mainly been in the form of books, book chapters, peer-reviewed articles, etc., but with little attention paid to managing scientific data, processes, and outputs. Essentially, each data item is of value in answering specific research questions, but there were no systems put in place to make it FAIR. Nonetheless, the re-use and proper management of research data are becoming increasingly important as donors move toward the global trend of lodging digital public goods (DPGs) and publications in the public domain [10]. DPGs are open-source software, open data, open standards, open AI models, and open content that conform to privacy and other best practices and standards, international and domestic laws, and do no harm [10,11].

To solve these challenges, this study proposes an IS that will centralize data management. The IS tightly couples the following major software components, namely, common ontology [12], data management plan [13], data collection tools [14–16], and the data warehouse [17]. These coupled software and data resources are centrally accessible [18]. The platforms are expected to manage legacy data, digitize data collection, clean and create analytics-ready data, champion open data, showcase open science, etc. These are geared toward supporting the *icipe* vision of being a global pioneer research institute in insect science whose vision is to improve the well-being of humanity and the environment through

innovative and applied research, impact assessment, evaluation, and sustainable capacity building. These approaches can be adopted widely for scientific data management

## 2. Literature Review

This section will give a brief introduction to the research data, project life cycle and thereafter discuss literature related to research data management.

### 2.1. Project Lifecycle

Research data are mainly generated by project activities. Figure 1 shows the different stages of a data-driven project lifecycle. They are discussed below:

- Proposal development and submission: This involves doing a write-up that is normally submitted to a donor for funding consideration. Generally, it entails outlining a research problem, proposed methodology, budget, and resources involved.
- Project commissioning: This marks the beginning of the project after the donor accepts and funds a research project.
- Data collection: This is the process of fetching raw facts about a research problem. People employ digital tools to collect data (e.g., filling out digital forms on digital platforms, such as smartphones) while others use manual processes (e.g., filling out printed forms) when collecting data.
- Data analysis: Data are analyzed using different mathematical formulas (e.g., regressions,and Bayesian analysis) to extract insights from the data. The insights provide leads to solving the research problem and inform decision-makers and policy formulators.
- Data archiving and sharing: The raw data, processed data, metadata data, and other related information are stored for re-use. The data should be stored in a format that makes it easy to access, share, and be consumed by other systems, e.g., text file, csv, jpeg, and png.
- Project decommissioning: This marks the end of project activities. The donor is given a report on successes and challenges.

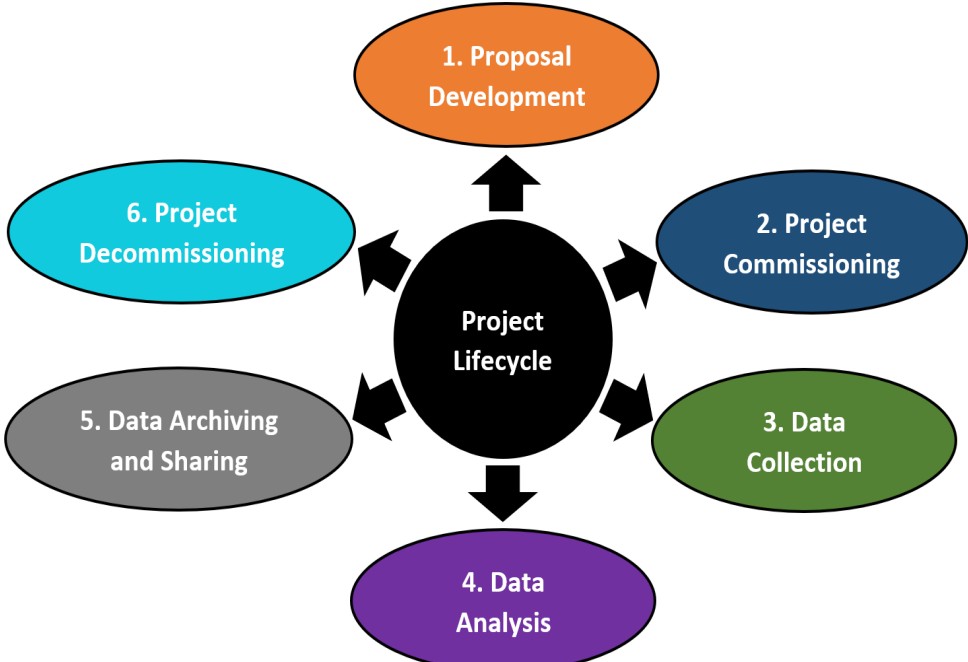

**Figure 1.** Project life cycle.

*2.2. Data Management Aspects*

Technological transformation has shifted data collection from traditional paper-based methods to digital data collection. The traditional paper-based method is costly with delayed data turnaround time, and is prone to human errors. In digital data collection, electronic devices, such as tablets or smartphones, are embraced in collecting, organizing, and sharing data. The digital devices enable users to collect accurate data in a faster, more efficient, and more cost-effective way [19]. Such devices are called the internet of things (IoT) since they are able to transmit the collected data to sink nodes in the internet [20]. For example, smartphones have a set of in-built sensors (such as position, motion, and GPS) that are explored to maximize the data collection experience, and fetch accurate data [21]. However, smartphones come with limited computing (disk storage, processing, and random access memory) capabilities that should be considered when planning data collection.

The International Livestock Research Institute (ILRI) has a set of data management software. Data collection is built on the open data kit, while the data warehouse is built on the Comprehensive Knowledge Archive Network (CKAN) and Dataverse. Their data are described with metadata and controlled vocabularies so that they are easy to discover and interpret. Data can be exported in an open file format, such as CSV, XML, GeoTiff, etc. [22]. Due to privacy reasons, personal data are filtered out before the public domain. However, some data are private and require authorization to access. Their data resources are licensed under a creative commons license [23]. It is noted that some raw data should be organized into a standard MS Excel sheet, then archived in the data warehouse. Their data comply with the FAIR principles of data management [22]. However, it is not stated how controlled vocabularies are managed. This research was not able to see how their software resources, outputs and tools were interlinked [24].

Luis et al. [25] developed a service-oriented architecture for the integrated management and analysis of multi-omics and biomedical imaging data. The Java web-based system was structured to meet the FAIR principles of data management and metadata. This facilitated storage and analysis of raw data and metadata from various omics, microscopy, and biomedical imaging modalities in an integrative manner with the ability to accept metadata queries from web-based and scientific applications. Proof-of-concept and use cases from plant biology and clinical studies were performed. The approaches were able to tackle the complexities and the ever-increasing volume of omics and biomedical. These consequently allow multi-modal data management, high throughput, and the generation of large and highly multi-dimensional datasets in life science [25].

The agricultural sector is characterized by a widespread use of different data formats, a tight connection to specific hardware implementations, and a lack of interoperability standards. Jacob et al. [26] addressed the potential of applying publicly available information sources to optimize crop production in Denmark. Free and publicly available data (polygons, satellite images, topographic maps, and orthophotos) were aggregated using GeoNode, an open-source web-based software (i.e., data infrastructure) that allowed information to be queried, merged with other datasets, and analyzed. GeoNode had Geo-Explorer to provide geographical information systems (GIS) data on a web application, GeoServer to do data management, and PostgreSQL as a database. GeoNode was configured on a virtual server and could be scaled horizontally depending on demand for increasing processing power and/or memory. Users could upload and share data using standard formats (Shape or GeoTIFF) for central management. GeoNode has standard protocols, such as web map services and web feature services, that enable users (including 3rd parties) to be assigned rights to access specific data items. They evaluated their data infrastructure based on the ease with which stakeholders could access (visualize, download) data that have a universal format to ensure interoperability and the ability to perform analysis on existing data. The data infrastructure enabled them to explore high-quality, freely available data and open possibilities for performing further analysis and improving crop production [26].

In the Philippines, cassava phytoplasma disease (CPD) is a major threat to cassava farming. Irma et al. [27] built a drone-based GIS solution for detecting CPD and informing farmers and relevant stakeholders so that they take the necessary steps to prevent the spread of the disease. On the raw data, they studied features such as color intensity, pixel intensity, ratio and coordinates plotting, and featured layer. The system was evaluated by its ability to detect infected stalks of cassava with CPD [27].

Saikanth et al. [28] performed a study to comprehend how farmers in the Nagarkurnool district of Telangana utilized the Agriculture Resources Information System Network (AGRISNET). They randomly selected and interviewed one farmer from the 120 villages, i.e., 120 respondents. They found that 46.67% of the farmers were using the system. From that 40.83% of checked quality of inputs information, 37.50% were interested in obtaining government agricultural schemes information, 28.33% needed crop protection information, 26.66 looked for market prices and 26.66% weather information. These results informed the authors of the popular service the system provided [28].

## 3. The Tightly Coupled Information System for Research Data Management

So far, Section 1 gives an introduction to research data management and the challenges faced by *icipe* and many other organizations. Section 2 has outlined literature related to the project life cycle, research data, and relevant data management strategies. This section proposes, designs, and implements a tightly coupled IS to support data management, i.e., the *icipe* research data management and archiving information system (iRDMA-IS). We believe this study's methodological approaches in realizing the iRDMA-IS will not only support the *icipe* data management activities, but will also inform organizations facing similar research or scientific data management challenges and can be adopted as a solution.

Figure 2 shows the proposed master IS architecture. Generally, the IS architecture is decomposed into independent and logical software components (SCs), which have different software specifications and architectural styles. Each SC has specific implementation details that are abstracted from each other but communicate to each other through well-defined application programming interfaces (APIs), i.e., component-based software engineering (CBSE). The SCs are, namely, common ontology (Section 3.2.1), data management plan (Section 3.2.2), legacy data management (Section 3.2.3), digital data collectors (Section 3.2.4), data warehouse (Section 3.2.5), and software version control (Section 3.2.6). Each component was tested independently, and bugs were cleared appropriately. Each SC has well-defined APIs that capture data or information as inputs from users or other SCs. The APIs also act as output channels to provide (processed) information to other SCs. Technically, each SC is a web platform hosted on a virtual machine on *icipe's* Ms. Azure virtual private network (VPN), and communication happens across the VPN cyberspace using secure hyper-text transfer protocol (HTTP) API (i.e., REST [29] and SOAP [30].) architectures. After interconnecting the tightly coupled SCs, we performed end-to-end integration testing to ascertain and meet the work flow envisioned in Figure 2. Each SC is maintained and is expected to evolve independently. In the maintenance and evolutionary process, we monitored and documented new and deprecated functions of each SC. If an SC is not maintainable, it can be replaced with another one in a plug-and-play fashion [31]. The selection of the SCs was guided by several factors that are described in Section 3.1. To use any service in the iRDMA-IS, a user fills out a service request form [32], which is posted to the systems administrator for approval and authorization.

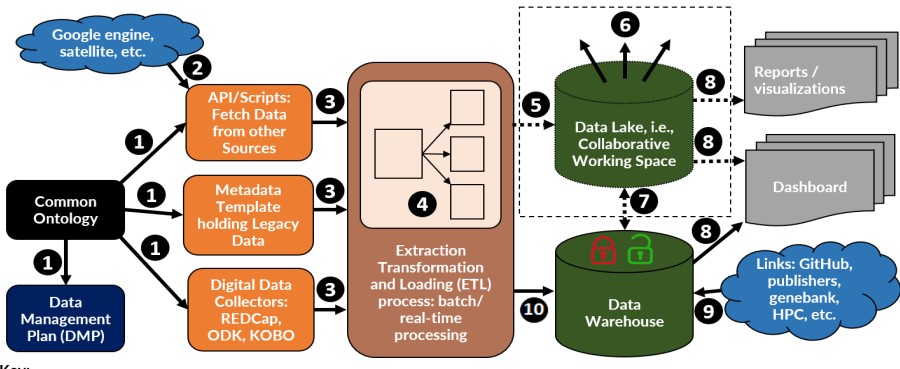

**Key:**
1. Common ontology annotated to variables in the DMP, new data from collectors, legacy data, and data from cloud systems
2. Fetch readily available data from other sources e.g. Google engine, satellite, etc.
3. Load raw data to memory for further cleaning
4. Extraction transformation and loading (ETL) processes: cleaning, profiling, and annotation; batch/real-time
5. Load clean and annotated raw data to the data lake for analysis and visualization [work in progress]
6. Build data analysis pipelines e.g. modeling and machine learning etc. [work in progress]
7. Execute APIs to archive meta-data, dictionary, protocols, experiment plan, raw data, and clean data [work in progress]
8. Execute APIs to summarize data and provide reports/visualization/dashboards [work in progress]
9. Data resources in the data warehouse updated with links to external data sources
10. Load clean and annotated raw data to the data warehouse for visualization and archiving

**Figure 2.** Data and software engineering pipelines.

### 3.1. Software Components Selection Criteria

We selected the different software components considering the following criteria:

- Open-source: The *icipe* research data management and archival (RDMA) policy advocates for open-source software tools and prefers to go for licensed software as a last resort; some donors assert the same directives.
- Software license: We were keen on selecting an open-source software that has a license that would allow us to copy and install it in our VPN/local premises, inspect the code and build upon it to meet our needs; licenses such as creative commons attributions [33] allow all these.
- Open-source community of software engineers around the product: Around the globe, there are many software engineers who are actively involved in extending and developing new products on pre-existing systems. Note that configuring open-source software components is also not a trivial task. Therefore, with a rallying community of software developers behind a software, we were at least guaranteed of obtaining quick responses/solutions when we needed support, and we could also re-use readily available and tested extensions that could support our user needs.
- Reduced software development time and cost: Developing software requires time and is quite costly. Ideally, ready-to-use open-source components with an open-source license reduce software development time and cost.
- Reliability: We selected components that were fully developed and tested. We also considered components that were pre-configured with security mechanisms to secure their APIs and abstracted layers.
- Plug-and-play, API, and replacement. The proposed master data infrastructure pipelines in Figure 2 were designed to have SCs that can be removed and replaced with efficient ones in a plug-and-play fashion. SCs with robust APIs guarantee communication between themselves.
- Maintenance and security: We selected SCs that are actively maintained by vendors who release regular updated SCs, which are efficient, stable, and secure. However, in the future, we will upgrade SCs after a thorough analysis to ascertain that the new functions are coherent with existing SCs and the intended user specifications and software functions.
- Software documentation: Most SCs come with detailed technical and user manuals. This was a point of reference when troubleshooting, integrating the SC with others, understanding their architecture and development.
- On-premise skills: There are quite a number of open-source software packages that provide similar functions but are written in different programming languages, e.g., [34,35].

We, therefore, settled on software tools that matched our software engineering skillset and ones that would not give us a steep learning curve before starting the actual configuration.

### 3.2. Coupling the Software Components, Their Specifications and Functions

The SCs are accessible and described (a snapshot is on Figure 3) on the website, https://dmmg.icipe.org/ (accessed on 1 September 2022). Their specific selection criteria, user requirements, function, customization, and coupling procedures are described in this section as follows.

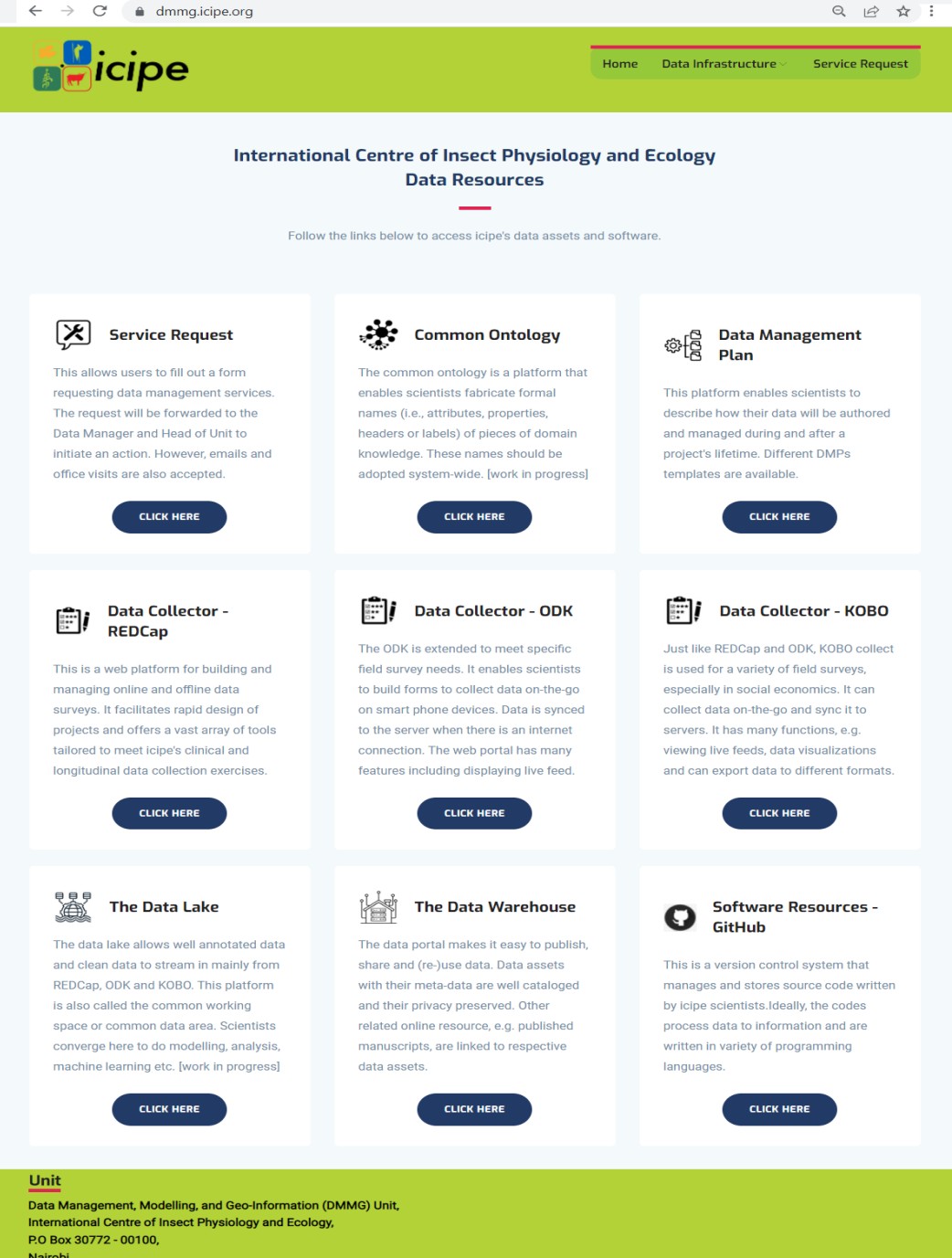

**Figure 3.** The landing web page to *icipe's* research data management and archiving infrastructure, the iRDMAI.

### 3.2.1. The Common Ontology

Ontology is the formal naming of data items that are representations of a domain of knowledge. Ontologies also describe the interrelationship of the data items. Ontologies are context-dependent projection models of reality. In computer programming, ontologies are identifiers. When retrieving data from the database when the identifiers are not standardized, quite a number of data items were left out since identifiers are named at the programmer's discretion. This was a bottleneck for machines and humans. For example, the date of birth as a data identifier could appear as DoB, date_of_birth, DateOfBirth, birth, birth_date, etc. In the human eye, these data identifiers have a similar meaning; however, for machines, they are all different. This made it difficult to holistically fetch certain pieces of data for purposes of analysis.

We worked around this challenge by having the analogy that data items should be aligned to standardized or similar vocabulary, i.e., common ontologies, in a process called ontology alignment. Ontology alignment is the idea of formally naming similar pieces of domain knowledge to have similar identifiers, attributes, properties, headers, or labels and noting their interrelationships. This makes it easy for humans and machines to fetch and share all common or similar data at once. At the same time, the data become machine interpretable when shared across different SCs in the iRDMA-IS since they are referenced and consumed with respect to similar identifiers.

Just like machine learning models, common ontologies are built for specific domain knowledge, e.g., wine [36], human disease ontology [37], gene ontology [38], malaria ontology [39], environment ontology [40], bio-collections ontology [41], etc. Most of these common ontologies run on free-to-use licenses, e.g., creative commons. We, therefore, did not re-invent the wheel by defining ontologies from scratch but rather re-used and normalized the already existing ontologies and then incrementally built on them in the domain of insect science. We note that, to the best of our knowledge, there is no ontology in insect science. At *icipe*, ontologies alignment is performed by domain experts who want to use or re-use ontologies in their data management activities. Nonetheless, for each defined ontology, we append a standard set of metadata, e.g., identification, description, data created, domain, created by, knowledge domain/category, and interrelationships (to other classes or properties). We note that new ontologies are defined by domain experts, validated, and saved in the database.

We reviewed different software that manage ontologies as per the criteria defined in Section 3.1 and we settled on WebProtégé [42] since it has been widely used for ontology management [43–45]. We configured and hosted it on our VPN [12]. WebProtégé allows users to load ontologies but does not offer a function to merge them. This research merged several ontologies (e.g., malaria ontology [39], and environment ontology [40]) so that users could access them from one interface. This research also noted WebProtégé searching, adding, deleting, and editing ontologies and their metadata. However, it does not provide an export function for ontologies selected by the user. This study therefore extended the functionality of WebProtégé to enable users to select, preview, and export ontologies of their choice to a data dictionary. The exported standardized data dictionary can be integrated with legacy data, data from APIs, and data collectors as discussed in Section 3.2.4. The downloaded data dictionary can also be appended to a data management plan for proper data planning over the life cycle of a project, as discussed in Section 3.2.2.

### 3.2.2. The Data Management Plan (DMP)

A DMP describes how data will be managed before and after the completion of a project. Generally, a DMP outlines how the research project will collect, organize, analyze, store, and share data. A DMP is an integral part of an institution that works with research data. It assists in aligning its data with the FAIR principles of data management in the lifecycle of a project [46–48]. In light of these, a handful of donors targeted by *icipe* scientists provide a DMP template (in Microsoft Word, online forms, etc.) to be filled. The DMP is then submitted together with the proposal. Scientists at *icipe* seek alternative platforms

(e.g., [34,35]) to create DMPs. Over time, such documentation is scattered over several platforms, making it hard to keep track of those DMPs to keep a record or even re-use them; these are challenges facing project-based scientific research communities [46]. The same scientists do not fully trust those platforms in the sense that someone could access and use their work without their consent. At times, our scientists waste time looking for DMP templates and guides for specific donors.

Researchers across the globe have tried to solve this problem in different ways. For example, the Data Curation Center [49] configured an open-source web-based DMP platform (from [35]) to enable UK researchers to create DMPs instead of seeking the same service from other platforms. This can be adopted and used for use cases, e.g., engineering and manufacturing, to manage project-based research data when adopted at early stages of a project [46].

As per the criteria defined in Section 3.1, we reviewed various open-source DMPs and settled on Research Data Management Organizer (RDMO) [34]. RDMO is widely used by other organizations [46,48]. We reconfigured and hosted it on our virtual premises [13]. The platform is dedicated to serving *icipe* scientists and partner institutions only. We provide our scientists with login passwords to access the system and create DMPs.

We note that the DMP configured by the Data Curation Center [35] has DMP templates for many donors and funding agencies. However, by default, the RDMO has no DMP templates. We therefore had to incrementally integrate DMPs of various donors that our scientists target, e.g., Horizon2020 [50], and Wellcome [51], which was time consuming. A scientist then filled out a DMP template and exported it to either an editable (MS Word, Latex, Rich Text, Open Office, etc.) or non-editable (e.g., PDF) format. The RDMO system provides us with functions, such as snapshots (for saving various instances of a work in progress DMP), collaboration (scientists can invite other users to work on a DMP), task (to set milestones that should be accomplished in the DMP), user profile (to change their authentication detail), etc. We now have the capability of centrally managing our scientists' DMP needs.

We note that the hosted DMP [13] highly relies on the WebProtégé ontology management platform described in Section 3.2.1. In that, if a donor's DMP requires a scientist to state variables that will be considered for data collection, then the scientist will log into the common ontology, select, preview, and download the standardized variables. In the same fashion, we tailored a DMP to suit the data needs that all scientists should fill before submitting their proposals. This guides the data and grants the management office to plan ahead for possible resources a project might need when awarded a grant. Moreover, where applicable, the selected ontologies will feed into the data collection systems, which are discussed in Section 3.2.4.

### 3.2.3. The Legacy Data Management

As indicated in Section 1, *icipe* has numerous legacy data that are generated, project wise. Research projects are commissioned only when they are funded and decommissioned at the end of project activities. We sampled a few decommissioned projects and analyzed the procedures of data collection, and how the data were organized and stored. We noted that most projects had metadata, raw data, and clean data. Some had a data dictionary and experimentation protocol. For us to fetch legacy data from scientists in an organized manner, we developed a standard MS Excel workbook [52] for documenting and organizing those data. The template, shown in Figure 4, has the following worksheets: metadata, protocol, data dictionary, raw data, and clean data.

The metadata worksheet captures the general information about the project, e.g., the titles, description, PI, PI email, collaborators, donor, and start and end dates. The protocol worksheet is filled with the research's experimentation protocol, including the design of structures for how the experiments were set up as a means by which data were collected or the results generated. The data dictionary worksheet defines variables used to capture data in the raw and actual worksheets. The raw and actual data are populated in the raw and

actual data worksheets, respectively; the data headers should be defined and referenced from [12]. Images, videos, etc., might not conform to this style but can be organized in folders to bring out the same meaning.

| | A | B | C |
|---|---|---|---|
| 1 | Element | Required/Optional | Provide information here |
| 2 | Title | Required | |
| 3 | Description | Required | |
| 4 | Name of principal investigator | Required | |
| 5 | Email of principle investigator | Required | |
| 6 | Collaborators (i.e. list of researchers involved) | Required | |
| 7 | Theme | Required | |
| 8 | Key words | Required | |
| 9 | Donor/funding agency | Required | |
| 10 | Start date of project | Required | |
| 11 | End date of project | Required | |
| 12 | Region e.g. East Africa, West Africa, Asia, Europe etc. | Required | |
| 13 | Country(ies) | Required | |
| 14 | Administrative area(s) e.g. Kakamega County, Suba District etc | Required | |
| 15 | Name of contact person | Required | |
| 16 | Email of contact person | Required | |
| 17 | Date uploaded | Required | |
| 18 | Citation narrative | Required | |
| 19 | Is this third party data? | Required | |
| 20 | Upload third party proprietary agreement, if applicable | Optional | |
| 21 | Acknowledgment statement | Required | |
| 22 | Article(s) published; link | Optional | |
| 23 | | | |
| 24 | | | |

Metadata  Dictionary or Vocabulary  Protocol  Raw-Data  Clean-Data

**Figure 4.** Meta-data template MS Excel workbook.

### 3.2.4. The Digital Data Collectors

This study chose digital data collection compared to manually filling out paper questionnaires, which is labor intensive, prone to errors, and expensive [19]. We identified three web-based data collection software packages using the criteria in Section 3.1. The web versions are namely: open data kit (ODK) [15], KoBoToolbox [16] and REDCap [14]. They are hosted in our VPN. ODK [53] and KoBotoolbox [54] have Android mobile versions only, while REDCap has both REDCap [55] for Android and REDCap [56] for iOS. The Android and iOS mobile applications are downloaded from Play and the App Stores and installed on the respective Android and iOS-enabled devices. These data collectors are meant for different research data collection needs. For instance, REDCap fits well with longitudinal and clinical surveys. REDCap runs on a commercial license, while ODK and KoBoToolbox are open source. KoBoToolbox is built on top of ODK and has advanced features (e.g., export functions, and data visualization on interactive maps) compared to ODK. Surveys for social and economic data are supported by both ODK and KoBoToolbox. However, this study customized ODK to meet project-specific needs, while KoBoToolbox is used institution wide.

For the data collectors, several projects can be defined. Within each project, several forms can be defined. Within each form, several labels (questions) and controls to capture data can be defined. Technically, the system administrator receives a form (in MS Word, PDF, etc.) from the project PI to digitize and make it ready for digital data collection. He/she then consults the common ontology management platform [12] where he/she will select and download the appropriate ontologies. Then fuse them against each data capture control on the digital form.

Depending on a question's flow of logic, different control flows can be defined, e.g., conditions and skip logic. Validations can be set against each question in the form to ensure specific rules for data items are met (e.g., numbers and emails) before a form is saved. These are some quality control checks that can be put in place to improve data quality. We note that standardizing data variables, i.e., ontology alignment, improved data quality and made it easy to fetch data that has a variable name from the entire database Mazandu et al. [43].

Moreover, all the selected digital data collectors are able to collect a wide range of types of data, e.g., location, image, video, and audio. After the form is digitized and the PI has reviewed and tested it, the mobile version of the respective digital data collector is installed

on smartphones and the digital form(s) are loaded. We note that other devices, such as the GPSMAP 64s [57], are used to capture accurate geographical positions accurately, then fed back to the mobile data collector where applicable.

On the web version, appropriate permissions (e.g., add, view submission, edit, and delete submission) and roles (e.g., administrator, project manager, project viewer, and data collector) are assigned to a user against a specific form that is deployed. A deployed form is identified by a quick response (QR) code. A QR code is made up of a two-dimensional black and white pixel pattern; it is a two-dimensional version of the bar code [58]. A user uses a mobile device to scan and validate the QR code. The mobile device that scans the QR code is validated, and the form design stored on that device is ready to collect data on the go. This improves data security.

During data collection, data are securely stored on the device. When there is an internet connection, either through data or Wi-Fi services, the user selects the saved data and uploads them to the server. A user can also opt to set uploads to be automatic as soon as the form is saved and there is an internet connection. The data collector can edit or delete data before uploading them; once uploaded, the data can only be edited or deleted from the web version. On the web versions, data can be previewed and downloaded. KoBoToolbox and REDCap offer more advanced (e.g., previews on interactive maps, and histograms) and export functions (e.g., to SPSS, KML) compared to ODK. REDCap offers even more advanced functions, e.g., data analytics, export vocabulary, and dynamic SQL.

Data can be exported from web platforms and given to users to start data analysis. The data can be extracted programmically and pre-processed further to meet specific system or user needs using custom extracted, transformed, and loaded (ETL) scripts. The ETL scripts are built on the respective data collector APIs, i.e., ODK [59], KoBoToolbox [60] and REDCap [61]. In the ETL scripts, raw data are fetched from the digital data collection systems, transformed (i.e., various quality control checks, e.g., common ontologies, outliers, missing values, and consistency), and loaded into the data warehouse (to be discussed in Section 3.2.5) or exported to formats ready to be consumed by processing platforms, e.g., STATA. Thereafter, the data are the back up in the VPN systems for future reference since those are the primary/raw data.

Nonetheless, our scientists wrote pieces of script to fetch data from various data sources, for example, weather data from the Earth Resources Observation and Science Center [62], Shuttle Radar Topography Mission [63], and pest distribution data from [64–66]. The data and metadata were fetched, re-used, and correlated with primary or secondary data (collected from the laboratory, fieldwork activities, etc.) to discover new knowledge [67–69]. It is therefore imperative that the fetched metadata are aligned with ontologies defined in the common ontology management platform (discussed in Section 3.2.1) before they are fused with existing data in the respective data collector using through executing inbuilt APIs in ODK [59], KoBoToolbox [61] and REDCap [60].

### 3.2.5. The Data Warehouse

Using the software selection criteria defined in Section 3.1, we selected, installed, and configured the comprehensive knowledge archive network (CKAN) web-based and open-source data warehouse system on our VPN to meet our data needs [17]. CKAN is a robust system for data archiving and has been adopted by various institutions and governments, such as the New South Wales State of Australia [70], United States Government [71], the International Livestock Research Institute [72], etc.

The system has various in-built mechanisms for data archiving on the web. The datastore stores structured data (e.g., CSV and spreadsheets) and a user can access the data using its simple web API or queries. The filestore stores whole (unstructured data) files (e.g., CSV, spreadsheet, image, and video) in the file system. Data in the file store cannot be queried but can be accessed using appropriate APIs calls [73,74]. The web service gateway interface and NGINX are used for securely running and hosting the system, Sorl and Jetty for searching information in the system, etc. [75]. There are also free-to-use extensions built

by third-party software engineers that enable various functions such as download counters, connecting to an active directory for authentication, integrating geographical information systems base maps, etc. [76].

Generally, the institution's organogram [77] indicates that research programs are managed under themes, while support units work independently but support all the themes. The programs and support units run projects that generate data. This study customized the default data warehouse to suit this organogram workflow. Furthermore, in order to ensure that data are well organized in the data warehouse, this study proposed that the system administrator be the only user who can authorize data uploading into the data warehouse. We note that data and respective metadata (outlined in Figure 4) can be uploaded directly or can be bundled and loaded into the data warehouse by calling appropriate APIs [78]. Ordinarily, legacy data are uploaded directly into the data warehouse, while closed project data in the digital data collectors are uploaded by executing appropriate APIs, e.g., ckan.logic.action.create.package_create(), from the digital data collectors discussed in Section 3.2.4. Due to data privacy and ethics issues, project metadata and its data can be made public or private; the PI provides appropriate consent. Public data are licensed under the creative commons attribution [33].

Software source codes used in processing research data, especially during the analysis stage, are uploaded to the version control system (discussed in Section 3.2.6) and made public or private. The links to the source codes are updated against the respective data items in the data warehouse. Links to the respective published articles are also updated against the data resources.

This study adopted Google Analytics to build dashboards from user activities. It collects, summarizes, and stores various user analytics of all our web-based systems in the iRDMAI, and we can design dashboards with different content that targets specific user (e.g., data manager and software engineer) needs. Visualizations and dashboards are tailored and integrated into articles to be published and reports.

We developed dashboards in Google Analytics [79] for us to understand user behavior and activities on the data warehouse. We built dashboards using live feeds of specific key performance indicators, e.g., count of downloaded data, number of users, and device used to access the system (e.g., mobile phones or computers). These make us understand different metrics and build strategies to optimize the system and support our users in a better way. For example, a lot of downloads of a certain dataset could tell us that the dataset could be of high quality, leading to a review and advising other users to do the same. In essence, this is business intelligence.

The concept of open data, open science, and reproducible science is realized in the data warehouse since people around the globe can easily access the raw data, data vocabulary, experimentation protocol, methodological steps (e.g., scripts) in processing that data, processed data, articles published, and any other relevant information. Consequently, anyone can fetch the well-described data, go through the methodology, and reproduce the same results.

Moreover, the FAIR principles of data management are also realized. That is, data are findable through the search function, facilitated by Sorl and Jetty software. Archived data in public mode are easily accessible on the web-based system. However, for private data, users must request the resources; after approval, a download link is sent to the user by email. Data are interoperable since the archived data can be downloaded in various formats that are consumable by other applications, e.g., CSV, jpeg, tif, xls, and xlsx. The raw and actual data are re-usable since they are archived and well described with metadata, data vocabulary, and experimentation protocols. This makes the data understandable and re-usable.

The concept of the common data model is also realized since data streams into the data warehouse from the digital data collectors (discussed in Section 3.2.4) and legacy management (discussed in Section 3.2.3). As previously discussed, the respective data

resources were fused with common ontologies at those stages. Consequently, a user can fetch pieces of similar or common data variables to perform a holistic analysis.

### 3.2.6. Version Control Systems

Considering the software selection criteria outlined in Section 3.1, we settled on GitHub as the platform to store source codes of various software (e.g., DMP, and common ontology) and pieces of software code. GitHub is open source and web based. The process of saving code into a GitHub account is called commit. GitHub enables our software engineers and scientists to collaborate when writing code. After they commit their code, the changes made can be tracked. If the changes are not satisfactory, the user can roll back to the previous version or roll forward if the immediate version was fine. Code stored on another GitHub account can be forked into our account; this allows us to centralize code that was already saved in another account. The forked source code can be modified (where necessary) without affecting the original source code. However, the modified forked code can be updated on the original source code by sending a pull request to the original source code owner who pulls the modifications and merges them into the original source code. The code can also be made public or private. Private source code is a work in progress or needs consent to be made public. The public source code data can be re-used and referenced appropriately [80].

### 3.3. Results and Discussion

We note that there are many performance metric evaluation strategies depending on the core functions of the IS. For instance, Saikanth et al. [28] built an IS for agricultural service delivery and evaluated it based on the number of users and specific services they accessed. Jacob et al. [26] developed a data infrastructure that enabled agricultural stakeholders to easily access and analyze GIS data and evaluated it based on its ease of accessibility and interoperability. We should note that the primary goal of this research was to design and couple platforms (i.e., iRDMA-IS) for collecting, collating, and promoting data re-use while adhering to the FAIR principles of data management and open data/science. Based on the data we collected from the time the software components were deployed, we considered evaluating the iRDMA-IS based on its accessibility, interoperability, new and returning users, the number of datasets archived, and the cumulative frequency of users who have accessed the software components.

We will discuss results for data collectors (REDCap and ODK) and the data warehouse from 1 August 2021 to 31 August 2022. We recently deployed the common ontology, data management plan, and KoBoToolbox data collector; we will analyze 2 months' results, from 1 July 2022 to 31 August 2022. These results were automatically captured and relayed to Google Analytics [79]. We fetched those data and built various visualizations using Matplotlib [81].

### 3.3.1. Accessibility

Figures 5–7 show the geographical distribution of users globally, while Figures 8–11 show a bar graph of the percentage distribution of users against different countries. We note that at least 259, 180, 1,334, 68, 52, and 31 people accessed the data warehouse, ODK, REDCap, Common Ontology, DMP, and KoBoToolbox respectively, across the globe. The Common Ontology, DMP, and KoBoToolbox have relatively few numbers since we relied on 2 months of data. Generally, from these results, Kenya had the highest number of people accessing the software components. *icipe* is based in Kenya, and a big percentage of its staff and research activities are based there. That could be the reason why we have a higher percentage of people accessing the systems. Nonetheless, the distribution is also dominated by people in Europe, the United States, the United Kingdom, and East Africa. These could be users who we partner with and collaborate with on different scientific activities. Journal reviewers also demand access to data and methodology (e.g., computer programs, and scripts) when reviewing scientific papers. Some of our scientists publish their data on

the data warehouse, meaning the (international) reviewers are part of these statistics. We see that iRDMA-IS has gained global traction, making *icipe* scientific data and relevant resources available globally.

We also note that the data in the warehouse are open (but licensed under Creative Commons Attribution [33] license) and well described with relevant metadata, such as the name and email of the principal investigator, collaborators, the project start and end dates, etc. Furthermore, most, if not all, datasets include a data dictionary that explains the variables and relevant information under study. Where applicable, each dataset has an experimental protocol attached describing how experiments were structured to fetch data. Other relevant information, such as published papers, technical reports, etc., is also attached. As an example, [82], out of many other projects (that encapsulate their respective metadata and other detailed data resources) was archived in the data warehouse, meaning anyone can access these data resources and reproduce the experiments. These articulate the findability, accessibility, and reproducibility concepts, which are emphasized in the FAIR principles of data management, open science, and open data. All these are realized in this research to showcase *icipe* and partner organizations' scientific data resources.

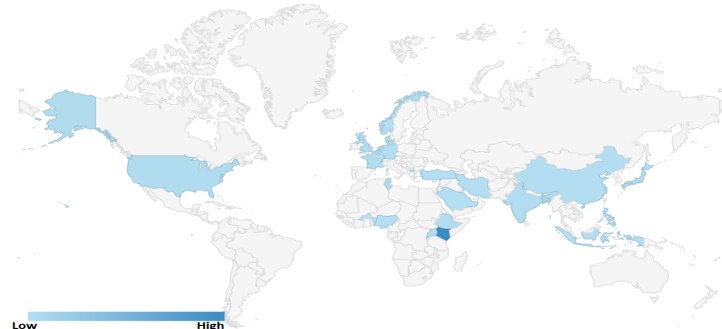

**Figure 5.** Data warehouse—distribution of users globally.

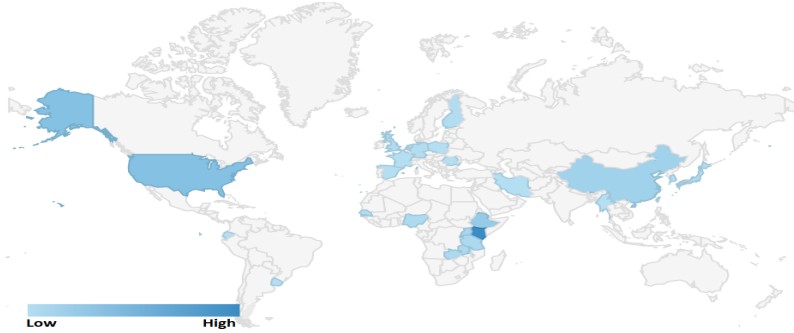

**Figure 6.** ODK—distribution of users globally.

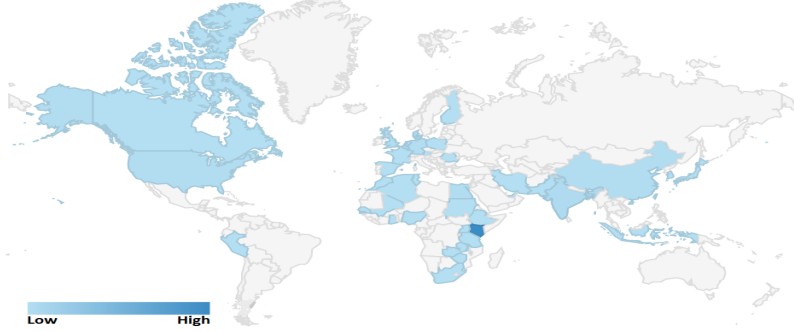

**Figure 7.** REDCap—distribution of users globally.

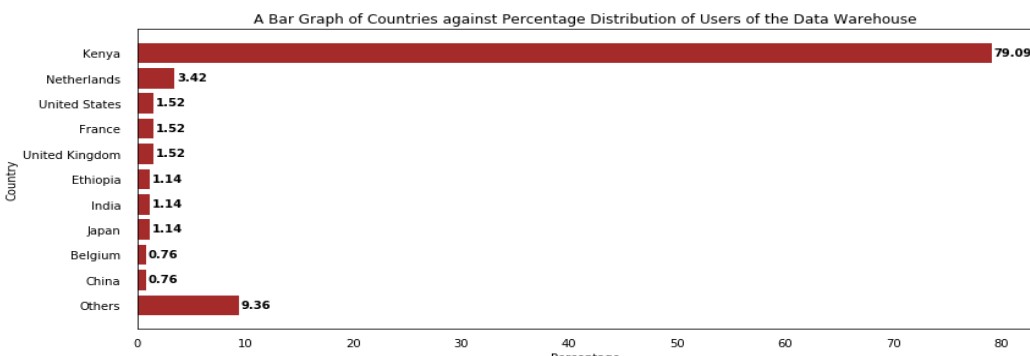

**Figure 8.** Data warehouse—percentage distribution of users by country.

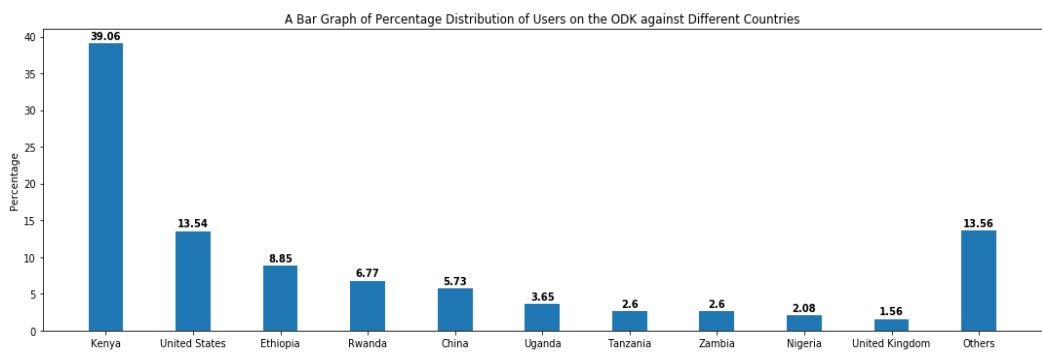

**Figure 9.** ODK—percentage distribution of users by country.

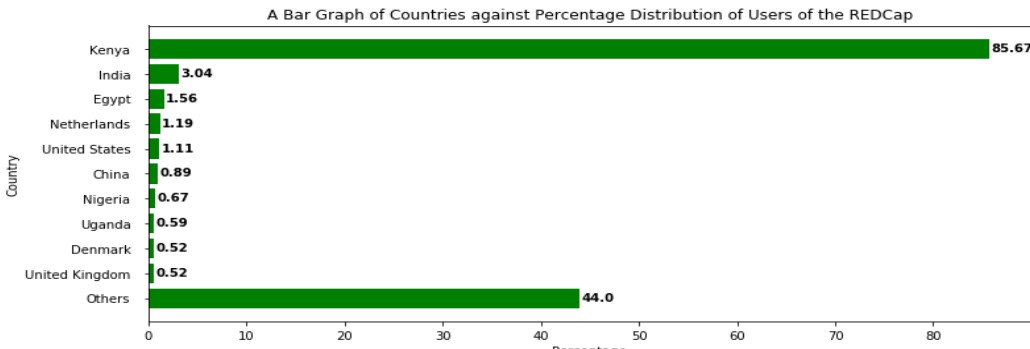

**Figure 10.** REDCap—percentage distribution of users by country.

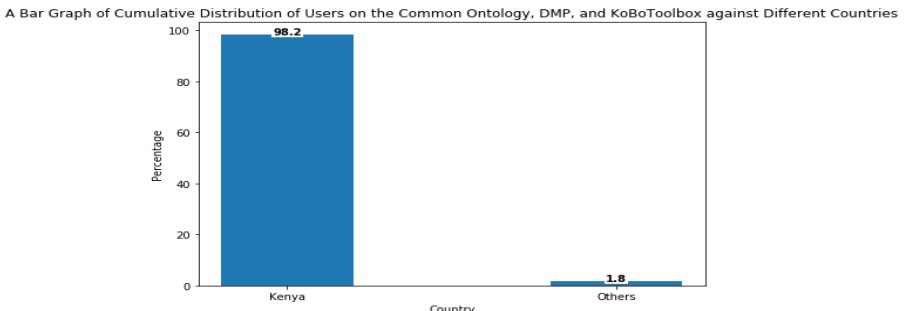

**Figure 11.** Cumulative distribution of users on the common ontology, DMP, and KoBoToolbox against different countries.

3.3.2. Web Hits, New and Returning Users

Figure 12 shows the share of the total number of web hits for the various software components. The total number of web hits were 20,228. A total of 60%, 35%, and 4% came from the REDCap, data warehouse, and ODK, respectively. The remaining 1% was from common ontology, DMP, and KoBoToolbox; this is significantly low since we only analyzed two months' results compared to the 13 months' results of the other software components. We suppose REDCap had more hits since it handled more projects that require data collection services compared to ODK. In Section 3.2.4, we noted REDCap handles mostly clinical longitudinal studies, while ODK does social economics studies. We note that both REDCap and ODK require authentication to access their content. In the future, we need to distinguish between web hits on the index and other web pages. Nonetheless, since REDCap handles clinical data, most likely, most of the *icipe* work is clinical. The data warehouse performs the archiving of data, meaning 36% of web hits were from users who were pursing archived data/information services.

Figure 13 illustrates new and returning visitors to the data warehouse, ODK, and REDCap. New visitors are those who have never accessed the software component before, while returning visitors are those who have. Google uses cookies to manage this service. We note that the data warehouse had the highest number of new users, followed by REDCAP and then ODK. When put together, the common ontology, DMP, and KoBoToolbox had the highest number of returning users; this could be due to the same users (i.e., software engineers, and data designers) who returned to access the systems, and their percentage distribution is shown in Figure 14. The highest number of new users in the data warehouse could have been as a result of new users who sought their data to be deposited, new journal reviewers, new users who were interested in various data services, etc. REDCap had more new users than ODK. In the previous paragraph, we saw REDCap handled more scientific data than ODK since projects have a life cycle (as stated in Section 2.1), and when a new project starts, most of the time new partners are engaged. This could have been attributed to it having more new users than ODK. Moreover, the data warehouse, REDCap and ODK had 7,143, 12,110, and 824 web hits, respectively. Cumulatively, the common ontology, DMP, and KoBoToolbox had a total of 151 web hits. From these, 56.3% were new users. Since projects have a life cycle and new users (partners, donors, etc.) come on board, so do new journal reviewers, etc. The 56.3% could indicate to us that science at *icipe* is vibrant.

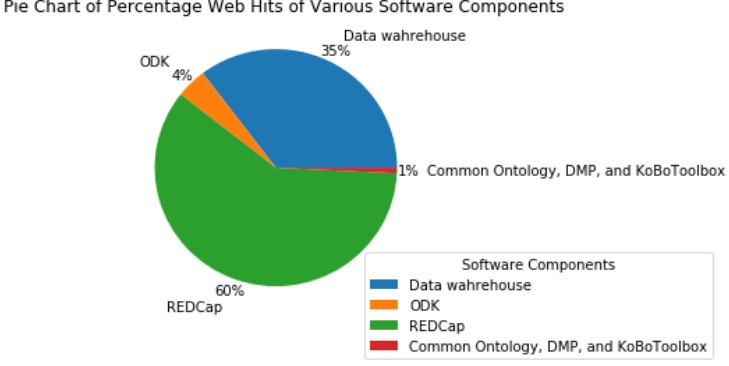

**Figure 12.** Percentage web hits of the various software components.

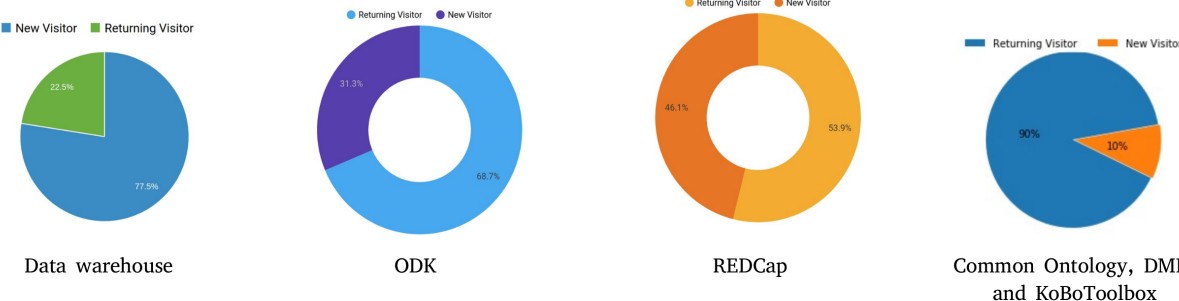

**Figure 13.** New and returning users on the various software components.

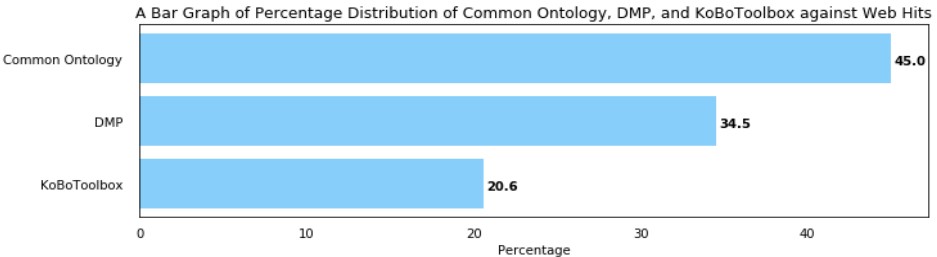

**Figure 14.** Common ontology, DMP, and KoBoToolbox against web hits.

### 3.3.3. Number of Project and Respective Datasets Archived

In Section 2.1, we stated that at *icipe*, research projects generate scientific data. Section 3.2.5 noted that each project has standard metadata that should be captured against it, but different datasets can be attached. As we work on uploading legacy (described in Section 3.2.3) data, this study uploaded all projects (without data) and metadata from the year 2000 to 2020. At the moment (September, 2022), as shown in Figure 15, we have a total of 259 projects and their metadata uploaded. From there, 17 projects have their respective datasets uploaded for the years 2020 and 2021. However, the *icipe* RDMA policy should be reinforced so that legacy data are collected, organized, and archived on the data warehouse so that the institution's scientific memory is reconstructed and centrally managed.

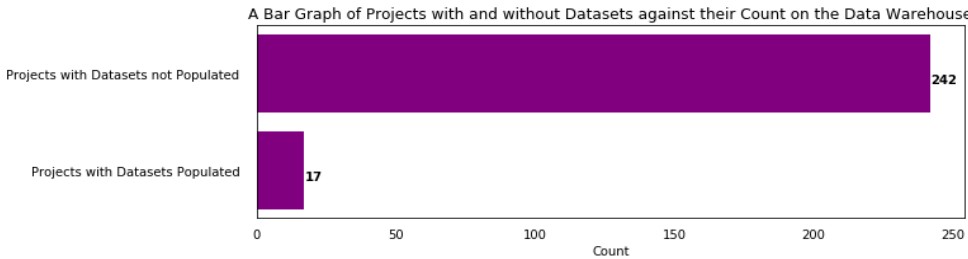

**Figure 15.** Data warehouse—projects with datasets populated.

### 3.3.4. Interoperability

For data to be interoperable, universal data formats for sharing data should be adopted. Sections 2.1 and 3.2.5 stated that most, if not all, scientists share their data with the public after they have exhausted their scientific investigations. It is at that point that the data (DPGs) are archived at the data warehouse for public access. The warehouse has some of its data stored in files with the following extensions: html, xls, doc, csv, tif, dbase, shp, shx, and asc. That means data shared with the public can be consumed by other applications and made reusable. Once again, these concepts of interoperability and reusability are emphasized in the FAIR principles of data management.

## 4. Conclusions

For more than 50 years of existence, *icipe* has never had a central data management system to handle the tons of data that it generates from various scientific research activities in entomology and related domains. Besides archiving digital data for re-use, fetching and holistic data analysis were also challenges. This study designed and implemented an information system to digitally collect, collate, and archive scientific data. Independent software components were selected, configured, interconnected, and installed in the institution's Ms. Azure's virtual private network. The coupled software formed the *icipe* research data management and archiving information system (iRDMA-IS). The information system emulates the common data model, where common ontologies are defined to align data variables in a system. In this case, data items in the data management plan, digital data collection tools, legacy data management, and the data warehouse were aligned with common variables to enable wholesome referencing and fetching of data. To request a service, users fill out a service request form and post it to the systems administrator, who initiates an appropriate support service. Generally, iRDMA-IS manages raw data, standardized data vocabulary, stores experimentation protocols, methodological processing procedures (e.g., scripts), actual data, links to published reports and articles, etc. At the data warehouse, data resources are set to public, if not private, to ensure data privacy, ethics, and security. Data can be public or private, but the principal investigator provides appropriate consent depending on the privacy, security, and ethical issues. From these, concepts of knowledge management, open science, open data, and reproducible science are realized. The iRDMAI-IS is currently used by the *icipe* scientific community and partner organizations. To the best of our knowledge, there is no such elaborate information system for research data management. The system is intended to revolutionize and digitize data management across the institution and act as a reference point for research data management. Generally, iRDMA-IS positions *icipe* and partner institutions in the global space of research institutions that practice the FAIR principles of data management, open data, open science, and reproducible science. These studies can inform institutions that face similar challenges.

## 5. Way Forward and Perspectives

In this study, we note that it is necessary to have a working area where users can be given rights to access specific datasets. In the future, we intend to integrate a data lake, which will sit between the digital data collectors and the data warehouse. On it, data will stream in from the digital data collectors. Users will then be assigned various permissions to do data analysis collaboratively. After they have exhausted their scientific activities, the data will be sent to the data warehouse for archiving.

Currently, the performance of our research activities is not tracked and monitored by well-established data-inspired learning mechanisms to elucidate their success or failure with possible recommendations. In future, we plan to create a "situation room" where the institution's research work will be captured, mapped, and visualized using graphs and numbers for purposes of strategic planning and decision making.

Most of the data we collect and feed into RDMA-IS come from farmers, and the scientific tools we develop should benefit them. However, we do not have a system to monitor, evaluate, and learn systematic pathways of bringing together multiple partners and, at the same time, enable them to access different technologies introduced to farmers, evaluate their impact, and track their adoption. Based on the lessons learned, we will be strategic in pushing forward adaptation, replication, and possibly up-scaling applied knowledge and technologies. In other words, we would like to establish a mega data, information, and knowledge management platform with features for scaling-up technologies, mapping beneficiaries, and monitoring their adoption.

**Author Contributions:** Conceptualization, K.S. and H.E.Z.T.; methodology, K.S. and H.E.Z.T.; formal analysis, K.S.; resources, H.E.Z.T.; data curation, K.S.; writing—original draft preparation, K.S.; writing—review and editing, K.S. and H.E.Z.T.; visualization, K.S.; supervision, H.E.Z.T.; project administration, K.S. and H.E.Z.T.; funding acquisition, H.E.Z.T. All authors have read and agreed to the published version of the manuscript.

**Funding:** The authors gratefully acknowledge the financial support for this research by the following organizations and agencies: the German Federal Ministry for Economic Cooperation and Development (BMZ), commissioned by the Deutsche Gesellschaft für Internationale Zusammenarbeit (GIZ) through the Fund International Agricultural Research (FIA), grant number: 18.7860.2-001.00; the Swedish International Development Cooperation Agency (Sida); the Swiss Agency for Development and Cooperation (SDC); the Federal Democratic Republic of Ethiopia; and the Government of the Republic of Kenya. The views expressed herein do not necessarily reflect the official opinion of the donors.

**Data Availability Statement:** The iRDMAI-IS and respective software components discussed on this manuscript are available on this link https://dmmg.icipe.org/.

**Conflicts of Interest:** The authors declare no conflict of interest. The funders had no role in the design of the study; in the collection, analyses, or interpretation of data; in the writing of the manuscript, or in the decision to publish the results.

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
