# Peer review of "A Novel Tightly Coupled Information System for Research Data Management"

_electronics, doi:10.3390/electronics11193196_

Round 1

Reviewer 1 Report

This work designed and implemented an information system architecture for research data management. It has a centralized platform for research data management. The software components are, namely: common ontology, data management plan, data collectors, and data repository. The information system aligned the institution’s scientific data to the FAIR data management principles and at the same time showcase open data, open science, and reproducible science. The goal is to adopt the proposed architecture to other organizations to manage research data. The paper introduces the software components in detail, such as, the common ontology, the data management plan, the legacy data management, the digital data collectors, the data warehouse, the version control systems. Overall, this paper is more like a technical report/description document other than an academic paper. Some comments are as follows.

1. The contribution of this work is to propose an information system that can be used to manage different research data. However, no experiment results can support this idea.

2. The authors introduce why they choose some software as a key component of the proposed architecture. However, the advantages of these software are not proved.

3. The English language and style are improvable. For example,

- Different forms of data collected such as laboratory/experiments notes, genomic data, household data, field notes and journals, photographs, geo-spatial, video and audiotapes, statistical package output files, technical reports, simulation, derived data, publications, etc. These data that is processed into information using various techniques that are relevant top the field of study.

Reviewer 2 Report

Thank you for the opportunity to review this interesting article. After reading it, I noticed the following aspects related to:

1. Abstract. The authors do not specify clearly what is the main objective of their research, the methodology used, the results or conclusions of their research. I suggest the authors to realize this.

2. Introduction. This section is brief and presents the influential factors in management data research of the authors, identifying a possible gap in the specialized literature.

3. Literature review. This section provides an introduction to research data, project life cycle and research data management. I suggest the authors to enrich this section with bibliographic notes because it is rather poor from this point of view.

4. The Tightly Coupled Information System for Research Data Management. The authors specify the following (line 148): As stated in Section ??, the technical aspects .... But what is the section?

Also in this section, the authors describe: Software Components Selection Criteria, Coupling the Software Components, their Specifications and Functions and make a small summary. The descriptions of the subsections are well done and documented and accompanied by eloquent graphics.

5. Conclusions. This section offers some ideas about the authors' own contributions.

Also, at the end, the authors propose several ways to continue improving their research (future research directions), but they do not present the limits offered by the current research. I suggest the authors to realize this.

Round 2

Reviewer 1 Report

In the revised manuscript, the authors carefully evaluated the iRDMA-IS based on accessibility, interoperability, new and returning users, the number of datasets archived, and the cumulative frequency of users who have accessed the software components. The results have proved that the proposed IS has a lot of users in different countries, and the Software Components have been used by many users. This kind of evaluation is more suitable for product introduction document.

The paper is overall well written. However, several key ingredients are still missing. An academic paper should prove the idea and support the contributions by comparative experiments, which means the proposed methods/systems should be compared with other state-of-the-art methods/systems. For an IS, the performance indicators may include system job throughput, system response time, mean time between failures, and etc.

The authors proposed an IS that will centralize data management in the institution. Apart from putting several software components together, it would be better if the authors can answer the following questions.

1. What is the biggest technical challenge to design an IS that will centralize data management in the institution? What performance indicators can be used to describe this challenge quantitatively?

2. Compare to other state-of-the-art ISs, how much have the performance indicators improved.

Round 3

Reviewer 1 Report

The reviewer still suggests the authors to test the performance of the proposed IS. Although the performance of the adopted software components has been tested by their designer, how does the whole IS perform is still unknown. Can the proposed IS reach the performance of the adopted software components?
